# Distribution of Extended-Spectrum-β-Lactamase-Producing Diarrheagenic *Escherichia coli* Clonal Complex 10 Isolates from Patients with Diarrhea in the Republic of Korea

**DOI:** 10.3390/antibiotics12111614

**Published:** 2023-11-10

**Authors:** Jungsun Park, Eunkyung Shin, Joohyun Han, Wooju Kang, Jaeil Yoo, Jung-Sik Yoo, Dong-Hyun Roh, Junyoung Kim

**Affiliations:** 1Division of Bacterial Diseases, Bureau of Infectious Disease Diagnosis Control, Korea Disease Control and Prevention Agency, Cheongju-si 28159, Republic of Koreaekshin1090@korea.kr (E.S.); jhan0212@korea.kr (J.H.); wooju0411@korea.kr (W.K.); knihyoo@korea.kr (J.Y.); 2Division of Antimicrobial Resistance Research, Center for Infectious Disease Research, Korea National Institute of Health, Cheongju-si 28159, Republic of Korea; 3Department of Biological Sciences and Biotechnology, Chungbuk National University, Cheongju-si 28644, Republic of Korea; dhroh@chungbuk.ac.kr

**Keywords:** ESBL, pathogenic *E. coli*, clonal complex, CTX-M

## Abstract

ESBL-producing *E. coli* is a public health concern in healthcare settings and the community. Between 2009 and 2018, a total of 187 ESBL-producing pathogenic *E. coli* isolates were identified, and clonal complex (CC) 10 was the predominant clone (*n* = 57). This study aimed to characterize the ESBL-producing pathogenic *E. coli* CC10 strains obtained from patients with diarrhea to improve our understanding of CC10 distribution in the Republic of Korea. A total of 57 CC10 strains were selected for comprehensive molecular characterization, including serotype identification, the analysis of antibiotic resistance genes, the investigation of genetic environments, the determination of plasmid profiles, and the assessment of genetic correlations among CC10 strains. Among the CC10 isolates, the most prevalent serotype was O25:H16 (*n* = 21, 38.9%), followed by O6:H16 (10, 19.6%). The most dominant ESBL genes were *bla*_CTX-M-15_ (*n* = 31, 55%) and *bla*_CTX-M-14_ (*n* = 15, 27%). Most *bla*_CTXM_ genes (*n* = 45, 82.5%) were located on plasmids, and these incompatibility groups were confirmed as IncB/O/K/Z, IncF, IncI1, and IncX1. The mobile elements located upstream and downstream mainly included *ISEcp1* (complete or incomplete) and *IS903* or *orf477*. Phylogenetic analysis showed that the CC10 strains were genetically diverse and spread among several distinct lineages. The results of this study show that ESBL-producing pathogenic *E. coli* CC10 has been consistently isolated, with CTX-M-15-producing *E. coli* O25:H16 isolates being the major type associated with the distribution of CC10 clones over the past decade. The identification of ESBL-producing pathogenic *E. coli* CC10 isolates underscores the possible emergence of resistant isolates with epidemic potential within this CC. As a result, continuous monitoring is essential to prevent the further dissemination of resistant ESBL-producing *E. coli* CC10 strains.

## 1. Introduction

Extended-spectrum cephalosporin resistance is a major threat worldwide, as cephalosporins are often used as first-line antimicrobial agents for treating infections caused by Gram-negative bacteria [1,2]. Cephalosporin-resistant bacteria produce enzymes called extended-spectrum beta-lactamases (ESBLs) [3].

In 2017, there were an estimated 197,400 cases of ESBL-producing *Enterobacteriaceae* among hospitalized patients and 9100 estimated deaths in the United States [1]. Among these *Enterobacteriaceae*, *E. coli* is one of the primary pathogens responsible for antimicrobial-resistant clinical infections [4]. The emergence of *E. coli* strains resistant to extended-spectrum cephalosporins was observed in the 2000s and has been continuously reported.

To understand the genetic diversity of *E. coli*, multilocus sequence typing has been most frequently used [5]. The prevalence of ESBL-producing *E. coli* has been reported, and sequence type (ST) 131 was the predominant sequence type [6]. Based on a study from Europe, the incidence of ESBL-producing *E. coli* ST131 was 20% in four European hospitals [7]. *E. coli* ST131 isolates have been reported in the Republic of Korea. Infection with ESBL-producing *E. coli* has significantly increased in community and healthcare settings, mostly due to the spread of ST131 *E. coli* clones [8,9,10]. Additionally, the presence of ESBL-producing *E. coli* ST131 in food animals has been detailed in previous literature [11].

Much of the literature focuses on the occurrence of clonally related antimicrobial-resistant bacteria, which is thought to contribute to understanding their transmission pathways. Therefore, it is necessary to analyze the clonal diversity of antimicrobial-resistant strains. The Korea Disease Control and Prevention Agency (KDCA) has collected and tested pathogenic *E. coli* isolates from patients with diarrhea to monitor antimicrobial resistance profiles. Between 2009 and 2018, a total of 187 third-generation cephalosporin-resistant pathogenic *E. coli* isolates were confirmed, and clonal complex (CC) 10 was the predominant clone. The objective of this study was to characterize ESBL-producing pathogenic *E. coli* CC10 strains obtained from diarrheal patients in recent decades to improve our understanding of CC10 distribution in the Republic of Korea and around the world.

## 2. Results

### 2.1. Isolation of CC10 Strains

Among the 4212 isolates, 295 *E. coli* strains were resistant to cefotaxime and ceftriaxone, and 187 out of the 295 strains were confirmed as pathogenic *E. coli.* The 187 pathogenic *E. coli* strains indicated that 102 carried the *eaeA* gene and were classified as EPEC; 44 strains carried the LT or ST genes and were confirmed as ETEC; and 41 strains carried the *aggR* gene and were identified as EAEC. Additionally, over half of the 187 strains showed resistance to two types of antibiotics: fluoroquinolone, tetracycline, aminoglycoside, macrolide or chloramphenicol. The sequence types (STs) and clonal complexes (CCs) of 187 resistant *E. coli* isolates were determined based on seven housekeeping genes (https://enterobase.warwick.ac.uk/species/ecoli/allele_st_search, accessed on 19 March 2022). The 187 resistant isolates were grouped into 77 STs. CC10 was the most prevalent clonal complex, comprising 57 (31.4%, 57/187) isolates with nine different STs (4, 10, 34, 218, 752, 1201, 1312, 1491, and 6955). A total of 57 CC10 strains were selected for the analysis of molecular characteristics.

### 2.2. Serotyping

The 57 pathogenic *E. coli* CC10 isolates belonged to 12 O serogroups and expressed eight different H antigens (Figure 1). The O serotypes were O25 (*n* = 22), O6 (*n* = 10), O101 (*n* = 5), O99 (*n* = 5) and O3 (*n* = 3). The more common H serotypes were H16 (*n* = 35), H33 (*n* = 6), H2 (*n* = 5), H10 (*n* = 5) and H30 (*n* = 3). The most prevalent serotype was O25:H16 (38.9%, 21/57), followed by O6:H16 (19.6%, 10/57), O99:H10 (9.8%, 5/57) and O101:H33 (9.8%, 5/57) (Figure 1). In silico FimH typing revealed 11 types of FimH. Of all CC10 strains, 16 (31.4%), 9 (17.7%), 8 (15.7%), 5 (9.8%), 4 (7.8%) and 1 (1.9%) were positive for FimH198, FimH54, FimH23, FimH30, FimH24 and FimH1194, respectively (Figure 1).

### 2.3. Prevalence of Genomic Determinants of Antimicrobial Resistance

A total of 34 antimicrobial resistance genes/mutations were detected, including those leading to resistance to six classes of antimicrobial agents, including beta-lactams (eighteen genes), fluoroquinolones (two genes and four mutations), tetracyclines (one gene), aminoglycosides (five genes), macrolides (one gene) and chloramphenicol (three genes) (Figure 1). All CC10 isolates encoded 13 different *bla*_CTX-M_ genes and *bla*_TEM-1,-30,-106,-126,-135,-141,-206 and -207_ genes. Fifty-six isolates carried the *bla*_CTX-M-15_ gene (31, 55%), and the next most common genes were *bla*_CTX-M-14_ (15, 27%), *bla*_CTX-M-55_ (3, 5%), *bla*_CTX-M-3_ and bla_CTX-M-27_ (2, 4%). Fluoroquinolone resistance genes/mutations were present in all isolates, of which quinolone resistance-determining regions (QRDRs) with *gyrA* gene mutations and plasmid-mediated quinolone resistance (PMQR) genes (*qnrS1* and *qnrB4*) were found in 32 (62.7%) and 23 (45.1%) isolates, respectively. Mutations in the *gyrA* gene were observed at codons 83 and 87, producing the single-residue substitutions S83L, S83A, D87G and D87N. Multiple fluoroquinolone resistance-associated mutations were detected in two isolates, specifically double mutations in *gyrA* (S83L with D87G or D87N). Tetracycline resistance was identified in 22 (43.2%) isolates, and macrolide resistance was confirmed in 13 (25.5%) isolates with genotypes *tet(A)* and *mph(A)*. Aminoglycoside resistance genes, including *aac(3)-Iia*, *aac(6′)-Ib-cr*, *aac(3)-Iid*, *aph(3″)-Ib* and *aph(6)-Id*, were harbored by 10 isolates (19.6%). These isolates carried one or two resistance genes. The following chloramphenicol resistance genes were detected in five (9.8%) isolates: *catB3*, *floR* and *cmlA1*.

### 2.4. Transmission of Bla Gene

The characterization of *bla*_CTX-M_ plasmids was performed to better understand the horizontal transfer of *bla*_CTX-M_ using conjugation. Of all CC10 strains, 82.5% (45/57) were capable of horizontal transfer through conjugation. The confirmed genes were associated with the presence of *bla*_CTX-M-15_ in 29 isolates, *bla*_CTX-M-14_ in 10 isolates, *bla*_CTX-M-55_ in 3 isolates, *bla*_CTX-M-3_ in 2 isolates and *bla*_CTX-M-27_ in 1 isolate. The four different plasmid compatibility groups were confirmed as follows: 21 isolates carried an IncB/O/K/Z plasmid, 12 isolates carried an IncF-type plasmid, 8 isolates carried an IncI1 plasmid and 2 carried an IncX1 plasmid.

### 2.5. Analysis of the Regions Surrounding bla_CTX-M_ Genes

Four different structures (type I (10 isolates), type II (23 isolates), type III (3 isolates) and type IV (9 isolates)) were identified regarding the genetic elements of *bla*_CTX-M_ (Figure 2).

A type I genetic structure was found in 10 isolates producing *bla*_CTX-M_ with *ISEcp1*-*bla*_CTX-M_-*orf477* genetic structures. The type II genetic structure was most common and identified in 23 isolates. Analysis of the region flanking *bla*_CTX-M_ revealed an *orf477* downstream sequence with a spacer region between the inverted repeat (IR) sequences upstream of *ISEcp1*. These genetic structures belong to the CTX-M-I group, such as *bla*_CTX-M-3,-15,-55_ possessing isolates. However, a type III genetic structure was identified in three isolates that had a different genetic element flanking *bla*_CTX-M_ and *ISEcp1* upstream and downstream of *IS903* (IR-*ISEcp1*-*bla*_CTX-M_-*IS903*). Three isolates had a type IV genetic structure, with *ISEcp1*-*bla*_CTX-M_-*IS903* as a transposable element.

### 2.6. Whole-Genome SNPs-Based Phylogeny of CC10 

Phylogenetic analysis was performed with 265 genomes of *E. coli* strains belonging to the international CC10, and a whole-genome SNP phylogeny was generated using *E. coli* K12-MG1655 as a reference (Figure 3). To better understand the global population structure of *E. coli* CC10, we identified a genome alignment in which 14,260 SNPs were identified. Phylogeny analysis of 57 clinical pathogenic *E. coli* isolates in this study was used to identify several distinct lineages, including ST4, ST10, ST34, ST752, ST1312 and ST1491. There was no observed substantial clustering related to the location or time of sampling during this study period.

There was observed related to CC10 isolated in the Republic of Korea from 27 reference strains. ST1312 carrying-*bla*_CTX-M-15-1_ and ST1312 carrying-*bla*_CTX-M-15-2_, which were isolated from a river in Sweden in 2013, were closely related, with 100 to 109 SNP differences within these study strains (20112155 and 20170132) [12]. These isolates had the O25:H16 serotype and common resistance genes, including *sul1*, *dfrA14*, *tet(A)* and *bla*_CTX-M-15_.

Five isolates (20121544, 20180444, 20181364, 20181484, and 20181747) from this study were similar to isolates collected from patients in China between 2017 and 2019 (6 to 91 SNP differences); they also harbored *bla*_CTX-M-15_ or *bla*_CTX-M-14_ and had mutations in quinolone resistance-determining regions (E18090, E18013, E19033, E19059, E15052, E20033, E17113, E19010 or E17090) [13].

Five isolates (20123618, 20130502, 20140370, 20140384 and 20140777) from this study were related to isolates 1512689, 1545515 and 3-6-R5 from patients in the United Kingdom and Australia, with 39 to 195 SNP differences. One isolate (287717) from the United Kingdom in 2016 was genetically similar to two isolates from this study (20160275 and 20161813), with the same genetic determinants, plasmids and serotypes, and there were between 33 and 46 SNP differences [14]. Two clinical isolates from China in 2017 (CN-202-F, CN-263-D) carried the same antimicrobial resistance genes, mutations and plasmids as similar human isolates (20182244, 20180595, 20181199, 20182201, 20182274, 20183083 and 20182759) in this study (54 to 58 SNP differences).

## 3. Discussion

ESBL-producing pathogenic *E. coli* sequence types have been extremely genetically diverse in the past decade. During this period, CC10 was the most prominent type; of 57 isolates with nine different STs, 30 were ETEC isolates, 16 EPEC were isolates and 11 were EAEC isolates. In this study, we compared the CC10 strains of ESBL-producing clinical *E. coli* derived from humans to describe their characteristics.

Over half of the strains with ESBLs showed multidrug-resistant (MDR) occurrence. The resistance rate was highest to tetracycline, followed by nalidixic acid, azithromycin, ciprofloxacin and trimethoprim/sulfamethoxazole. MDR *E. coli* with ESBLs has become a serious problem in public health because of the dissemination of the ESBL genes; this problem has threatened the treatment of bacterial infections [15].

In CC10 isolates, the incidence of CTX-M was highest, at 98.4% of the total, and the most dominant ESBL gene was *bla*_CTX-M-15_ (56%, 32/57). The majority of CTX-M-15-producing isolates had common features in that they belonged to serotype O25:H16 (53%, 17/32). We also confirmed the consistent presence of the *bla*_CTX-M-15_ gene and observed the highest levels in 2018. Previous studies related to *bla*_CTX-M-15_-harboring *E. coli* reported in the Republic of Korea include strains isolated from raw vegetables and food animals [11,16]. This result suggests that *bla*_CTX-M_-_15_-producing *E. coli* may circulate among food, food animals and humans and might contribute to the acquisition of resistance. 

Additionally, our study highlights the fact that plasmid acquisition is probably an important mechanism for the dissemination of CTX-M-producing pathogenic *E. coli*. Resistance to third-generation cephalosporins is caused by the acquisition of ESBL genes, primarily the *bla*_CTX-M_ gene [17]. Conjugation experiments were performed to confirm the horizontal transmission of plasmid-borne *bla*_CTX-M_ genes, and such transfer was found in 82.5% (45/57) of CC10 isolates. This suggested that the high incidence of CC10 isolates is caused by the horizontal transfer of ESBL genes between bacteria, which seems to be the best method of transmission. The predominant genotypes of the plasmid-mediated *bla*_CTX-M_ gene were CTX-M-15 (64.4%, 29/45) and CTX-M-14 (22.2%, 10/45). The mobile elements located upstream and downstream of the *bla*_CTX-M_ gene mainly included *ISEcp1* (complete or incomplete) and *IS903* or *orf477*, respectively.

Comparison of the WGS-based population structure of the 57 CC10 strains from the pathogenic *E. coli* from patients with diarrhea that were sporadically isolated from 10 regions showed that the strains were clonally related, with less than 70 SNPs separating them. When compared with the dissemination of CC10 worldwide, five isolates were highly similar to ETEC isolates from diarrhea patients in China collected from 2017 to 2019 [13]. Among the five isolates, one isolate was collected in 2012, and the other four isolates were collected in 2018. SNP analysis of those isolates led to the identification of 6 to 91 SNPs, thus indicating a close relationship among the isolates, even though they were identified in distinct countries. Additionally, these isolates carried the same antimicrobial resistance genes and plasmids as the strains in this study. The isolates from China were *E. coli* isolated from patients, suggesting that spread via unknown sources could lead to the dissemination of the clone. These results revealed that circulating CC10 strains from the Republic of Korea, as well as in other countries, were genetically closely related, which suggests the expansion of global or endemic populations.

Based on these observations, CC10 may become the most important strain in the Republic of Korea. During the last decade, CC10 strains of ESBL-producing pathogenic *E. coli* have been steadily isolated, and the rate of isolation has almost doubled recently. ESBL-producing CC10 *E. coli* isolates have already been reported as CTX-M-producing CC10 *E. coli* clones were found in swine fecal samples between 2017 and 2020 [18]. This finding suggests that the CC10 clone is emerging as one of the important clones with third-generation cephalosporin resistance. Several previous studies supported the idea that *E. coli* CC10 has a predominant clonal group associated with extraintestinal disease in both animals and humans [19,20]. Some regional monitoring investigations from Italy, Spain and Portugal showed that CC10 strains from humans, birds and swine were associated with multiple CTX-M-type genes [21,22,23]. 

Pathogenic *E. coli* has emerged as a major cause of food- and water-borne diseases in the Republic of Korea. Currently, most of the data on ESBL-producing pathogenic *E. coli* CC10 are from studies conducted in the Republic of Korea and may reflect a local situation. Therefore, the necessary data for the national management of pathogenic *E. coli* infections and related infections were collected. Future studies surveying the spread of the CC10 clone will need a collection of isolates from the clinic, food, water and the environment.

## 4. Materials and Methods 

### 4.1. Isolation of CC10 Strains

During the study period (2008–2018), 4212 *E. coli* isolates were obtained by the national surveillance system (Enter-Net and Pulse-Net Korea). These isolates were collected from stool or rectal swabs from patients with gastrointestinal symptoms, including diarrhea, abdominal pain, vomiting, nausea and fever. 

An initial investigation was performed to select cefotaxime- and ceftriaxone-resistant pathogenic *E. coli*. All isolates were identified using real-time PCR (Kogene Biotech, Seoul, Republic of Korea) to detect specific virulence genes of pathogenic *E. coli* according to the manufacturer’s protocol. The isolates were assessed for antimicrobial susceptibility by a broth microdilution method using customized Sensititre KRCDC2F plates (Trek Diagnostic Systems, East Grinstead, West Sussex, UK). The sequence types (STs) and clonal complexes (CCs) of cefotaxime- and ceftriaxone-resistant pathogenic *E. coli* isolates were determined based on seven housekeeping genes (https://enterobase.warwick.ac.uk/species/ecoli/allele_st_search, accessed on 19 March 2022). 

### 4.2. Whole Genome Sequencing (WGS)

Genomic DNA was isolated using a Blood and Tissue kit (Qiagen, Stockach, Germany) according to the manufacturer’s protocol. The purified total DNA quality was measured using a NanoDrop 2000 spectrophotometer (Thermo Fisher, Waltham, MA, USA). The concentration was determined with a Qubit 4 fluorometer using a high-sensitivity kit (Invitrogen, Waltham, MA, USA). Library fragment lengths were assessed through the use of a Bioanalyzer TapeStation with a DNA 1000 kit (Agilent Technologies, Inc., Santa Clara, CA, USA). A paired-end sequencing library was constructed with an Illumina DNA prep kit (Illumina, San Diego, CA, USA) following the manufacturer’s protocol. Sequencing was performed using a 500-cycle (2 × 250-bp paired-end) MiSeq reagent kit version 2 with a MiSeq sequencer.

### 4.3. Data Analysis and Molecular Characterization 

Raw sequences generated by Illumina MiSeq were quality filtered using FastQC, with the average quality set at Q30. Raw reads from the Illumina sequencing were quality trimmed using CLC Genomics Workbench 22 (Qiagen, Hilden, Germany). Contigs of genomic sequences were assembled with a minimum size threshold of 200 bp using the de novo assembler in CLC genomic workbench 22 (Qiagen, Hilden, Germany). Assembled sequences were analyzed for antimicrobial resistance genes (ResFinder 4.1), plasmid replicon types (PlasmidFinder 2.1), serotypes (SerotypeFinder 2.0) and *fimH* and *fumC* (CHTyper 1.0) using web tools available from the Center for Genomic Epidemiology (CGE) (http://www.genomicepidemiology.org/ (accessed on 15 August 2023)) [24]. Single-nucleotide polymorphisms (SNPs) were identified using CSI phylogeny 1.4 (https://cge.cbs.dtu.dk/services/CSIPhylogeny/, accessed on 20 April 2022) by comparing the *E. coli* K-12 MG1655 (GenBank accession no. U00096) reference strain with 265 *E. coli* genomes of the same sequence type (CC10) retrieved from EnteroBase (listed in Appendix A). The selection of SNPs was performed using the default parameters in CSI Phylogeny, which included a minimum distance of 10 bp between SNPs, a minimum of 10% of the average depth, mapping quality above 25 and SNP quality above 30. All insertions and deletions (INDELs) were excluded.

### 4.4. Plasmid Transfer by Bacterial Conjugation

Strains with cefotaxime resistance were examined by conjugation experiments using azide-resistant *E. coli* J53 as the recipient strain to confirm the transmission capacity of the *bla* genes [25]. Transconjugants were selected on MacConkey agar plates (Difco, Detroit, MI, USA) supplemented with cefotaxime (1 mg/L) and sodium azide (100 mg/L). The acquisition of the *bla* gene was confirmed by PCR and sequencing analysis.

### 4.5. Nucleotide Sequence Accession Numbers

The whole-genome sequences of these strains were deposited with the NCBI Sequence Read Archive (SRA) under the Bio-Project PRJNA628558.

## 5. Conclusions

ESBL-producing pathogenic *E. coli* CC10 isolates have steadily increased over the past decade. Presently, ESBL-producing pathogenic *E. coli* CC10 may become the most important strain in the Republic of Korea. Among these isolates, CTX-M-15-producing pathogenic *E. coli* O25:H16 isolates were the major CC10 clone. There have been a few studies that address this issue in CC10 isolates both in the Republic of Korea and in other countries. The identification of CC10 isolates highlights the possible emergence of resistant isolates within this CC with epidemic potential. Therefore, continuous monitoring will be needed to prevent the additional spread of resistant ESBL-producing *E. coli* CC10 strains.

## Figures and Tables

**Figure 1 antibiotics-12-01614-f001:**
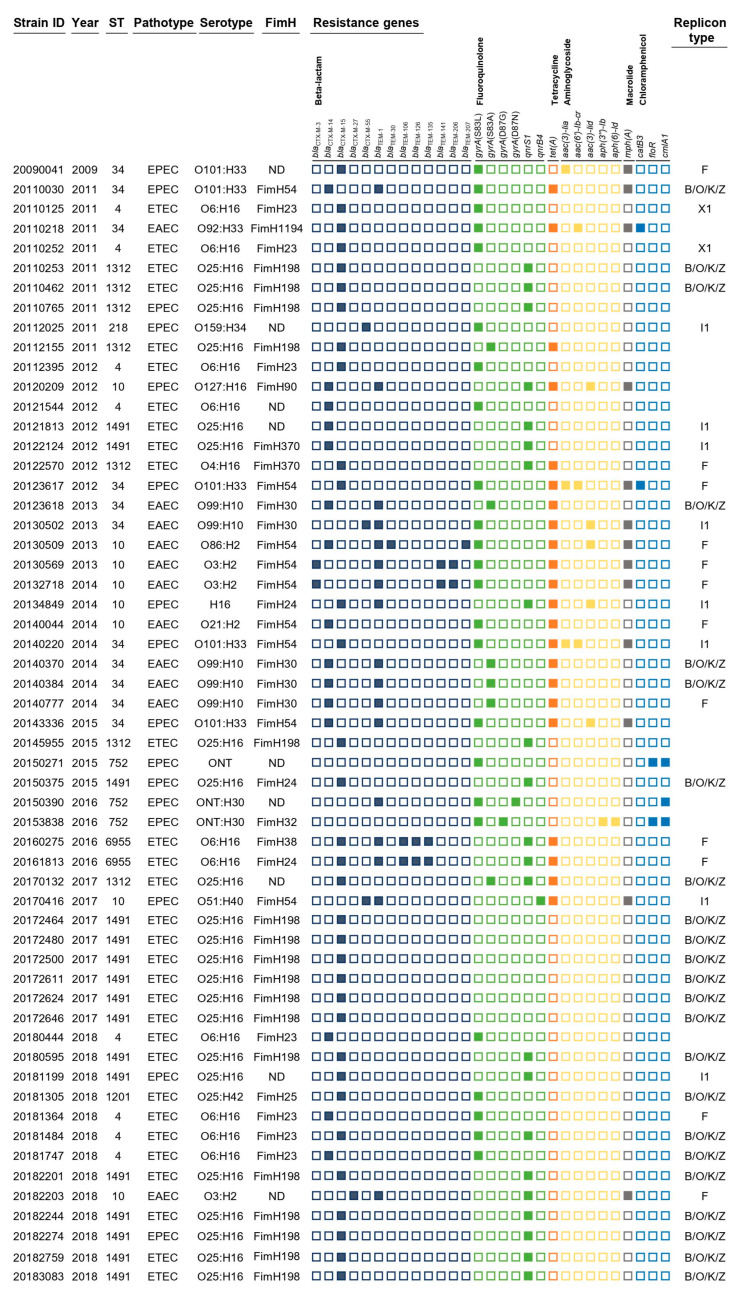
Characteristics of the 57 ESBL-producing pathogenic *E. coli* CC10 isolates in this study. The rows show sample ID and year of isolation, and columns represent ST, pathotype, serotype, antimicrobial resistance genes and plasmid replicon type. Antimicrobial resistance genes grouped by antibiotic class are demarcated by black lines under gene names.

**Figure 2 antibiotics-12-01614-f002:**
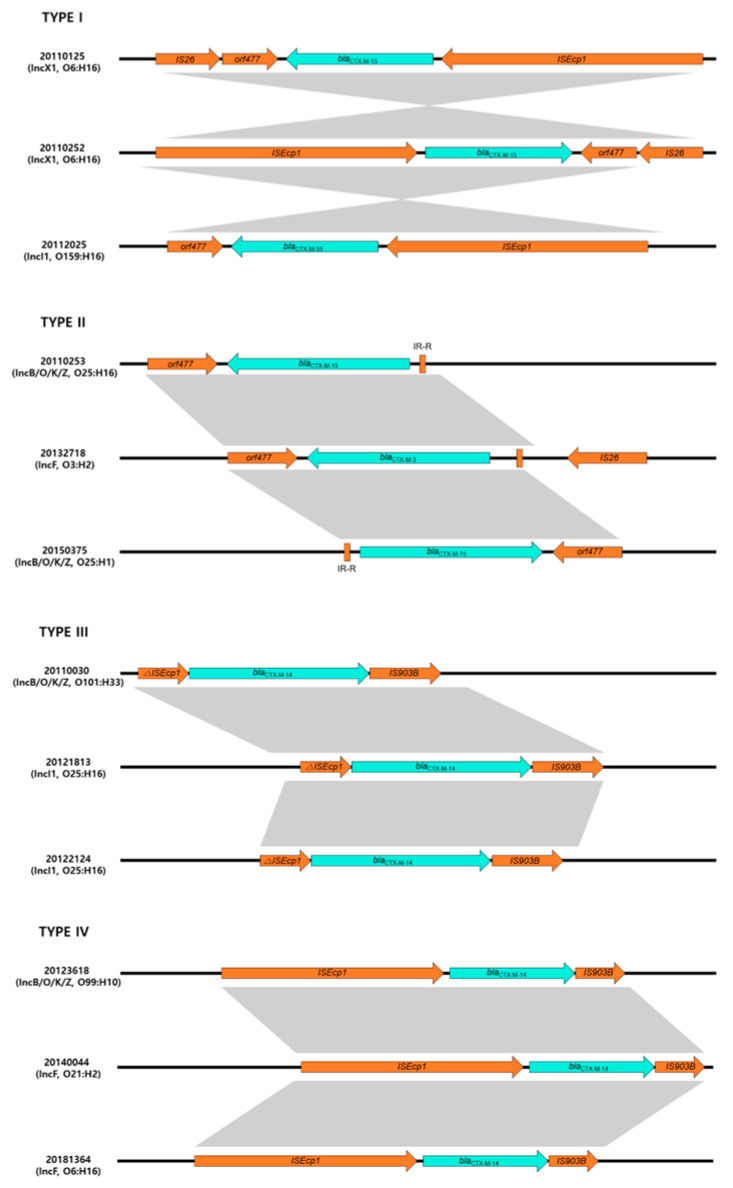
Genetic environment of *bla*_CTX-M_ gene in this study. Type I architecture (*ISEcp1*-*bla_CTX_*_-*M*-_-*ORF477*) was found in 10 isolates; type II architecture (*IR-R-bla_CTX_*_-*M*_-*ORF477*) was found in 23 isolates; type III architecture (*∆ISEcp1*-*bla_CTX_*_-*M*_-*IS903*) was found in 3 isolates; type IV architecture (*ISEcp1*-*bla_CTX_*_-*M*_-*IS903*) was found in 9 isolates.

**Figure 3 antibiotics-12-01614-f003:**
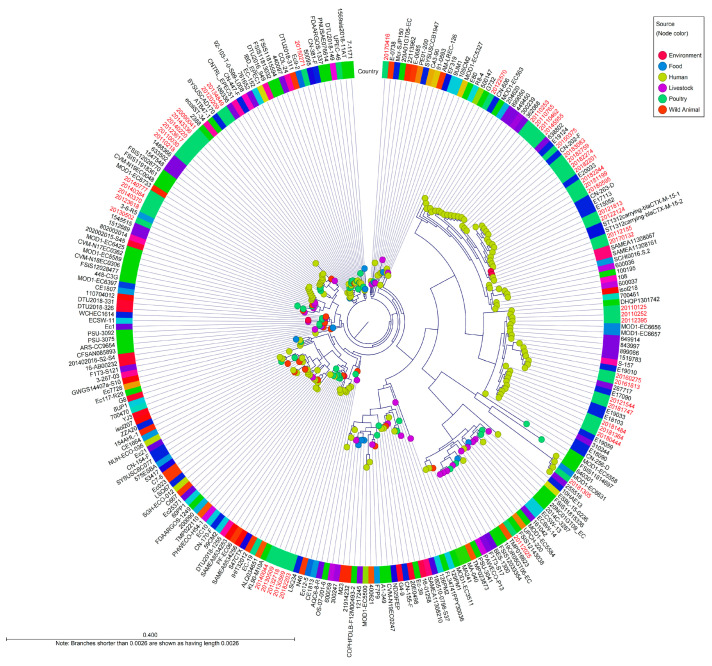
Whole-genome SNP-derived phylogenetic tree of CC10 isolates in a global context. The tree includes 265 international *E. coli* CC10 sequences, including the reference sequence of *E. coli* K-12 MG1655. The diagram depicts a phylogenetic tree with a genome alignment of the 14,260 SNPs for 265 global CC10 strains, including all publicly available isolates from EnteroBase. The node color represents the source of the isolates, and the colored ring around the tree indicates the country where the isolates were collected. Red colored names indicate the strains were isolated in the Republic of Korea. Isolate names colored black indicates that the strains isolated in other countries.

## Data Availability

All data generated for this study are contained within the article.

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
