# Peer review of "Distribution of Extended-Spectrum-β-Lactamase-Producing Diarrheagenic Escherichia coli Clonal Complex 10 Isolates from Patients with Diarrhea in the Republic of Korea"

_antibiotics, 2023, doi:10.3390/antibiotics12111614_

Round 1

Reviewer 1 Report

Comments and Suggestions for Authors

The authors describe the spread of ESBL-producing pathogenic E. coli in the Republic of Korea. Of particular concern, they noted, is the spread of strains from clonal complex 10, which could become epidemic. Therefore, the study was devoted to the analysis of ESBL isolates from CC10, their molecular characterization, and comparison of relatedness based on genomic SNIPs with CC10 strains described in enterobase.

My comments on the research an the presented paper are as follows:

1. pathogenic E. coli could be extraintestinal or intestinal. The reader should already know from the title that the article describes intetinal pathogenic strains

2. English language and sentence structure need improvement especially in the abstract section

3. the discussion should be rewritten

4. the authors are writing about plasmid profiles. I did not see any,  they probably mean incompatibility groups  (Inc) according  replication region analysis

5. describing the mechanism beta lactamases hydrolysis in the introduction is not necesarry 

6. ESBL - producing Enterobacteriaceae are not a rising problem, since they are already a huge  problem worlwide

7. the authors present in the introduction data ESBL and ST (131) data from Europe and the  United States, it would be more interesting to present more data from the Republic of Korea. 

8. did  the Korea Disease Control and Prevention Agency collected all pathogenic E. coli  in the Republic of Korea in the given time frame (2009-2018). 187 strains are a low number.

9. what is the distance between locations where the CC10 strains were isolated. Is there any connection? Are there any data about  food and environmental CC10 strains from the Republic Korea?

10 I think Figure 1 is not a real heatmap.  The letters are too small and therefore unreadable. The figure legend needs to be supplemented/changed.

11.    2.2 distribution in the subtitle should be deleted, or replaced with prevalence

12.  2.3 Transmission of bla genes: the authors state that  conjugation of bla genes was detected in 45 out of 57 strains. Given that they tested transconjugants, it would be interesting to know which bla genes were transferred.

Azide concentration 200 mg/L is very high. 

13. The whole-genome SNIP based phylogeny, and other data from the study as well as  considering the number of strains in Enterobase and their spread over time, I peronally think there is currently no major threat worldwide with regard to the spread of CC10. Hovever further studies should be performed in the Republic of Korea to exclude local, food/water and environmental associated outbreaks.

14. In my opinion, the article lacks key data for evaluating isolates of the CC10 complex and their association with infection and spread - the analysis of the presence of virulence factor geness. Since the authors have genomic sequences for all 57 isolates, I do not understand why these results, which could be informative, are not included in the paper.

15. the discussion needs to be rewritten and well structured, with a clear thread,   improved English and clear conclusions

Comments on the Quality of English Language

The English language and sentence style should be improved .

Author Response

  1. Pathogenic E. coli could be extraintestinal or intestinal. The reader should already know from the title that the article describes intestinal pathogenic strains.

- Response: We thank for the suggestions. This manuscript has described the intestinal pathogenic E. coli isolated from diarrheal patients. We have revised the title “Distribution of extended-spectrum-β-lactamase-producing pathogenic Escherichia coli clonal complex 10 isolates from patients with diarrhea in the Republic of Korea”

  1. English language and sentence structure need improvement especially in the abstract section

- Response: We thank for the suggestions. This manuscript has edited by the native English-speaking editors. We have improved this point in our revised manuscript and made additional modifications in response.

  1. the discussion should be rewritten

- Response: We thank the Reviewer for pointing this out and suggestions. We have revised the discussion section. We hope that these replies and modifications are satisfactory.

  1. the authors are writing about plasmid profiles. I did not see any, they probably mean incompatibility groups (Inc) according to replication region analysis

- Response: We thank the Reviewer for pointing this out and suggestions. We have revised this sentence to “Most blaCTX-M genes (n=45, 82.5%) were located on plasmids, and these incompatibility groups were confirmed as IncB/O/K/Z, IncF, IncI1 and IncX1.”

  1. describing the mechanism beta lactamases hydrolysis in the introduction is not necessary

- Response: As suggested by the Reviewer, we have deleted this sentence to “ESBL enzymes hydrolyze antibiotics, including penicillin and cephalosporins, making these drugs ineffective in treating infections.” in the revised manuscript.

  1. ESBL - producing Enterobacteriaceae are not a rising problem, since they are already a huge problem worldwide

- Response: As suggested by the Reviewer, we have revised this sentence to “ESBL-producing E. coli is a public health concern in healthcare settings and the community.”

  1. the authors present in the introduction data ESBL and ST (131) data from Europe and the United States, it would be more interesting to present more data from the Republic of Korea.

- Response: We thank the Reviewer for pointing this out and suggestions. We have added this sentence to “E. coli ST131 isolates have been reported in the Republic of Korea. Infection with ESBL-producing E. coli has significantly increased in community and healthcare settings, mostly due to the spread of ST131 E. coli clones [8–10]. Additionally, the presence of ESBL-producing E. coli ST131 in food animals has been detailed in previous literature [11].”

  1. did the Korea Disease Control and Prevention Agency collected all pathogenic E. coli in the Republic of Korea in the given time frame (2009-2018). 187 strains are a low number.

- Response: We thank the Reviewer for pointing this out and suggestions. During the study period (2008-2018), 4,212 E. coli isolates were obtained by the national surveillance system (Enter-Net and Pulse-Net Korea). These isolates were collected from stool or rectal swabs from patients with gastrointestinal symptoms, including diarrhea, abdominal pain, vomiting, nausea, and fever. All isolates were identified using real-time PCR (Kogene Biotech, Seoul, Korea) to detect specific virulence genes of pathogenic E. coli and assessed for antimicrobial susceptibility by a broth microdilution method using customized Sensititre KRCDC2F plates (Trek Diagnostic Systems, OH, USA). A total of 187 strains were confirmed, and EPEC constituted the highest proportion of strains (102 strains), followed by ETEC (44 strains) and EAEC (41 strains). We have added the following sentence in the revised manuscript in background section of materials and methods.

  1. What is the distance between locations where the CC10 strains were isolated. Is there any connection? Are there any data about food and environmental CC10 strains from the Republic Korea?

- Response: We thank the Reviewer for pointing this out. A total of 57 strains were isolated from 18 sporadic regions in Korea. We tried to find the relationship between these isolates, but it was challenging to find any correlation between the regions. We were also aware that comparing data about food or environmental strains from Korea was the best way to discuss these data. We tried to get data about food and environmental CC10 isolates from Korea, but unfortunately, we were not able to get the whole-genome sequences of CC10 isolates, including accurate information about the regions, isolate data, isolate source and sequence type. Also, there is literature about CC10 strains from Korea, but we could not find public whole-genome sequences. Therefore, we decided the best way to compare CC10 isolate sequence data is using that from humans, food or environments from around the world. The results showed that the ETEC isolates from diarrhea patients in China were closely related to our strains, with only 6 to 91 SNP differences, even though they were isolated from distinct countries. These data might suggest that these isolates could affect unknown sources and spread this clone. We hope that these replies and the modifications are satisfactory.  

10 I think Figure 1 is not a real heatmap. The letters are too small and therefore unreadable. The figure legend needs to be supplemented/changed.

- Response: As suggested by the Reviewer, we have replaced the figure with a high resolution. Also, we have revised/added the Figure 1 legend to “ Characteristics of 57 ESBL-producing pathogenic E. coli CC10 isolates in this study. The rows show sample ID and year of isolation and columns represent ST, pathotype, serotype, antimicrobials resistance genes and plasmid replicon type in this study. Antimicrobial resistance genes grouped by antibiotic class are demarcated by black lines under gene names.”.

  1. 2.2 distribution in the subtitle should be deleted, or replaced with prevalence

- Response: As suggested by the Reviewer, we have revised the subtitle 2.2 to “Prevalence of genomic determinants of antimicrobial resistance.”

  1. 2.3 Transmission of bla genes: the authors state that conjugation of bla genes was detected in 45 out of 57 strains. Given that they tested transconjugants, it would be interesting to know which bla genes were transferred. Azide concentration 200 mg/L is very high.

- Response: We thank the Reviewer for pointing this out and suggestions. We have added this sentence to “The confirmed genes were associated with the presence of blaCTX-M-15 in 29 isolates, blaCTX-M-14 in 10 isolates, blaCTX-M-55 in 3 isolates, blaCTX-M-3 in 2 isolates and blaCTX-M-27 in 1 isolate.” in the results sections.

We followed the transconjugant methods in much of the literature reported. Conjugation experiments were performed based on a previous study in which the sodium azide concentration was 100 mg/L [9]. We have revised the word “sodium azide (100 mg/L)” in the methods section. We hope that these replies and modifications are satisfactory.

  1. The whole-genome SNIP based phylogeny, and other data from the study as well as considering the number of strains in Enterobase and their spread over time, I personally think there is currently no major threat worldwide with regard to the spread of CC10. However further studies should be performed in the Republic of Korea to exclude local, food/water and environmental associated outbreaks.

- Response: We thank the Reviewer for pointing this out. The spread of ST131 ESBL-resistant E. coli clones has been reported in the literature. Before starting this study, we thought our data were similar to these results. We performed an experiment on pathogenic E. coli obtained from individuals with food- and water-borne diseases in Korea. A total of 4,212 E. coli isolates were collected, and 187 resistant pathogenic E. coli isolates were confirmed. We obtained a different result than we expected; in these isolates, CC10 was the most prevalent clonal complex, accounting for 57 (31.4%, 57/187) isolates. Additionally, the number of CC10 isolates has increased every year over the last decade. Therefore, we regarded the CC10 clone as one of the important clones in human clinical cases in Korea. ESBL-producing E. coli CC10 isolates have been reported in previous studies, and the predominant ESBL types were found to be CTX-M-55 and CC10 clones, which were also found in swine fecal samples between 2017 and 2020. In addition, blaCTX-M-15-harboring E. coli was also reported in raw vegetables and food animals in 2018. We tried to compare these sequence data; unfortunately, we could not obtain whole-genome sequences or accurate information, such as about regions, isolation data, isolate sources and sequence types. We hope that these replies and the modifications are satisfactory.

  1. In my opinion, the article lacks key data for evaluating isolates of the CC10 complex and their association with infection and spread - the analysis of the presence of virulence factor genes. Since the authors have genomic sequences for all 57 isolates, I do not understand why these results, which could be informative, are not included in the paper.

- Response: We thank the Reviewer for pointing this out. We knew that by comparing virulence factor genes, we could obtain better data. The 57 pathogenic E. coli isolates included 30 ETEC isolates, 16 EPEC isolates and 11 EAEC isolates. We tried to compare the virulence factors of CC10 isolates, and the genes were different depending on their pathotype. Our data are highly similar for each isolate depending on their pathotype. We have decided that whole-genome sequencing analysis was performed on all strains, but no difference was found in relation to region or isolation year. The results showed that 57 clinical isolates were genetically diverse and could be divided into eight distinct lineages based on O serotype, H serotype, FimH and genetic relationships with fewer than 70 SNPs separating them. We described this in discussion section. We hope that these replies and the modifications are satisfactory.

  1. the discussion needs to be rewritten and well structured, with a clear thread, improved English and clear conclusions

- Response: We thank the Reviewer for pointing this out and suggestions. We have revised the discussion section. We hope that these replies and modifications are satisfactory.

This concludes our response to the reviewer’s comments. We believe they have helped us improve our work considerably. We hope that they now find our manuscript acceptable for publication.

Reviewer 2 Report

Comments and Suggestions for Authors

In this manuscript, Jungsun et al. analyzed 57 third-generation cephalosporin-resistant E. coli isolates (CC10) by WGS, PCR and conjugation. The conclusion of this work is ESBL-producing pathogenic E. coli CC10 isolates are common in South Korea, and the continuous monitoring will be conducted to prevent further spread of resistant ESBL-producing E. coli CC10 strains. Please find the comments below from this reviewer.

This manuscript does not have the line number, and this makes it difficult to indicate the comments.

This manuscript will benefit the language polishing. For example, a revised version of the Abstract can be found below for the authors to consider:

“ESBL-producing E. coli represents a growing public health concern both in healthcare settings and the broader community. Between 2009 and 2018, a total of 187 ESBL-producing pathogenic E. coli isolates were confirmed, with clonal complex (CC) 10 being the predominant clone (n=57). This study aims to characterize ESBL-producing pathogenic E. coli CC10 strains obtained from patients with diarrhea, contributing to a better understanding of CC10 distribution in the Republic of Korea. These 57 CC10 strains were selected for comprehensive molecular characterization, including serotype identification, analysis of antibiotic resistance genes, investigation of genetic environments, determination of plasmid profiles, and assessment of genetic correlations among CC10 strains. Among the CC10 isolates, the most prevalent serotype was O25:H16, followed by O6:H16. ESBL genes were identified as blaCTX-M, with the most dominant ESBL genes being blaCTX-M-15 (56%) and blaCTX-M-14 (30%). Most blaCTX-M genes were located on plasmids, and these plasmid profiles were confirmed as IncB/O/K/Z, IncF, IncI1, and IncX1. The mobile elements located upstream and downstream mainly included ISEcp1 (complete or incomplete) and IS903 or orf477, respectively. Phylogenetic analysis revealed that the CC10 strains exhibited genetic diversity and were divided into several distinct lineages. In this study, it was observed that CC10 ESBL-producing pathogenic E. coli has been consistently isolated, particularly with CTX-M-15-producing E. coli O25:H16 isolates being the major type associated with the distribution of CC10 clones over the past decade. The identification of ESBL-producing pathogenic E. coli CC10 isolates underscores the potential for the emergence of resistant isolates with epidemic potential within this CC. As a result, continuous monitoring is essential to prevent further dissemination of resistant ESBL-producing E. coli CC10 strains.”

The title and the manuscript consider the E. coli isolates used in this study are pathogenic ones. How can you be sure they are pathogenic E. coli, not the commensal ones? This is an important question which should be well addressed.

Also, regarding these isolates, the authors need to find out how these isolates were obtained by the national surveillance system (Enter-Net and Pulse-Net Korea) between 2009 and 2018? Were non-enrichment media or selective media supplemented with antibiotics used to obtain these isolates? The background information will enable us to understand if conclusion in this study will well represent the AMR burden in South Korea.

In Abstract, it is expected to provide details and number for the readers. For example,

 O25:H16 (n=?, %), followed by O6:H16 (n=? %);  Most blaCTX-M genes (n=, 5);

Abstract: E. coli should be italicized, and please check the whole manuscript;

Page-2:

this sentence is confusing: ESBL-producing Enterobacteriaceae are rising a major public problem in healthcare settings and the community;

Please modify as: 197,400 cases of ESBL-producing Enterobacteriaceae among diversity of E. coli,

Do not start a sentence with a number: 57 CC10 strains 

Each figure should come with a figure legend: Figure 1. Heatmap shows sample ID, year of isolation, ST, Pathotype, Serotype, AMR profile and plasmid replicon type.

Page-7:

This sentence needs a revision: There was observed substantial clustering

Page 8

multidrug-resistant (MDR): MDR means the bacteria are resistant to drugs at least from three categories. Please review if this statement is correct.

Page-9:

Please modify as: and these 187 resistance isolates;

Author Response

  1. This manuscript will benefit the language polishing. For example, a revised version of the Abstract can be found below for the authors to consider:

- Response: We would like to thank the Reviewer for your constructive suggestions and comments on our work, which are helpful in improving the manuscript. We have applied the abstract section as you mentioned.

  1. The title and the manuscript consider the E. coli isolates used in this study are pathogenic ones. How can you be sure they are pathogenic E. coli, not the commensal ones? This is an important question which should be well addressed.

- Response: We fully thank the Reviewer for pointing this out and suggestions. During the study period (2008-2018), 4,212 E. coli isolates were obtained by the national surveillance system (Enter-Net and Pulse-Net Korea). These isolates were collected from stool or rectal swabs from patients with gastrointestinal symptoms, including diarrhea, abdominal pain, vomiting, nausea, and fever. All isolates were identified using real-time PCR (Kogene Biotech, Seoul, Korea) to detect specific virulence genes of pathogenic E. coli and assessed for antimicrobial susceptibility by a broth microdilution method using customized Sensititre KRCDC2F plates (Trek Diagnostic Systems, OH, USA). A total of 187 strains were confirmed, and EPEC constituted the highest proportion of strains (102 strains), followed by ETEC (44 strains) and EAEC (41 strains). We have added the following sentence in the revised manuscript in background section of materials and methods.

  1. In Abstract, it is expected to provide details and number for the readers. For example, O25:H16 (n=?, %), followed by O6:H16 (n=? %); Most blaCTX-M genes (n=, 5);

- Response: We fully agree with the reviewer. As suggested, we have revised this sentence to “Among the CC10 isolates, the most prevalent serotype was O25:H16 (n=21, 38.9%), followed by O6:H16 (10, 19.6%). The most dominant ESBL genes were blaCTX-M-15 (n=31, 55%) and blaCTX-M-14 (n=15, 27%). Most blaCTX-M genes (n=45, 82.5%) were located on plasmids, and these incompatibility groups were confirmed as IncB/O/K/Z, IncF, IncI1 and IncX1.” in abstract section.

  1. Abstract: E. coli should be italicized, and please check the whole manuscript;

- Response: We thank to the reviewer for noticing this error, we have replaced the word “E. coli” in italic type.

  1. Page-2: this sentence is confusing: ESBL-producing Enterobacteriaceae are rising a major public problem in healthcare settings and the community; Please modify as: 197,400 cases of ESBL-producing Enterobacteriaceae among diversity of E. coli,

- Response: We thank the Reviewer for pointing this out and suggestions. We have revised this sentence to “In 2017, there were an estimated 197,400 cases of ESBL-producing Enterobacteriaceae among hospitalized patients and 9,100 estimated deaths in the United States [1]. Among these Enterobacteriaceae, E. coli is one of the primary pathogens responsible for antimicrobial-resistant clinical infections [4].”

  1. Do not start a sentence with a number: 57 CC10 strains

- Response: We thank the Reviewer for pointing this out and suggestions. We have revised this world to “A total of 57 CC10 strains…” in revised manuscript.

  1. Each figure should come with a figure legend: Figure 1. Heatmap shows sample ID, year of isolation, ST, Pathotype, Serotype, AMR profile and plasmid replicon type.

- Response: As suggested by the Reviewer, we have added this sentence to “The rows show sample ID and year of isolation, and columns represent ST, pathotype, serotype, antimicrobial resistance genes and plasmid replicon type. Antimicrobial resistance genes grouped by antibiotic class are demarcated by black lines under gene names.”

  1. Page-7: This sentence needs a revision: There was observed substantial clustering

- Response: As suggested by the Reviewer, we have added this sentence to “There was observed related to CC10 strains isolated in the Republic of Korea from 27 reference strains.”

  1. Page 8: multidrug-resistant (MDR): MDR means the bacteria are resistant to drugs at least from three categories. Please review if this statement is correct.

- Response: We thank the Reviewer for pointing this out and suggestions. As you mentioned, multidrug-resistant is defined as resistant to at least three types of antibiotics. We have revised the sentence to “Over half of strains with ESBLs showed multidrug-resistant (MDR) occurrence.”

  1. Page-9: Please modify as: and these 187 resistance isolates;

- Response: As suggested by the Reviewer, we have revised this sentence to “and these 187 resistance isolates …”.

This concludes our response to the reviewer’s comments. We believe they have helped us improve our work considerably. We hope that they now find our manuscript acceptable for publication.

Round 2

Reviewer 2 Report

Comments and Suggestions for Authors

This reviewer appreciates the efforts of the authors to address my questions. Please further clarify: the terminology you need here is right for MDR? The MDR-isolates in this work were found to be resistant to at least one drug of three classes of the drugs? In, addition, in the methods, you describe the way to detect virulence genes etc, then, in the result, you need to provide the results: all isolates were found to be pathogenic E. coli or not.

Author Response

- Response: We thank the Reviewer for pointing this out and suggestions. A total of 4,212 E. coli strains were obtained by the national surveillance system during 2008-2018. Among 4,212 isolates, 295 E. coli strains were resistant to cefotaxime and ceftriaxone and 187 strains out of 295 were confirmed as pathogenic E. coli. The 187 pathogenic E. coli strains indicated that 102 carried the eaeA gene and were classified as EPEC; 44 strains carried the LT or ST genes and were confirmed as ETEC; 41 strains carried the aggR gene and were identified as EAEC. Additionally, over half of 187 strains have resistance to two types of antibiotics: fluoroquinolone, tetracycline, aminoglycoside, macrolide and chloramphenicol. we have added the sentence to in background section of material and methods.

This concludes our response to the reviewer’s comments. We believe they have helped us improve our work considerably. We hope that they now find our manuscript acceptable for publication.